

# Does quality of novice hand-tied square knots improve with repetition during a single training session?

Karen Tobias[1], Pierre-Yves Mulon[2], Alec Daniels[1] and Xiaocun Sun[3]

[1] Small Animal Clinical Sciences, University of Tennessee, Knoxville, Knoxville, TN, United States of America
[2] Department of Large Animal Clinical Sciences, University of Tennessee, Knoxville, Knoxville, TN, United States of America
[3] Office of Information Technology, University of Tennessee Knoxville, Knoxville, TN, United States of America

## ABSTRACT

**Background.** Knot tying is a key surgical skill for novices, and repetition over several training sessions improves knot tying. This study examined the effects of repetition within a single training session on quality of knotted loops and compared results of novice trainees and experienced surgeons.

**Methods.** Using 0.55 mm nylon monofilament fishing line, novices and surgeons each hand-tied 20 knotted loops, using a 2=1=1=1 configuration (surgeon's throw and three square throws). Loops were mechanically tested with a single load to failure.

**Results.** All loops tolerated five newtons (N) preload. More than 70% of novice and surgeon knots failed by slipping or untying, and 8.8% of novice knots and 2.5% of surgeon's knots were considered dangerous. Surgeons' loops had less extension at preload, indicating better loop security. However, during single test to failure, there was no difference in mean extension or maximum load between surgeons and novices. There was no significant difference in results of mechanical testing or modes of failure for the first and last ten knotted loops, or the first, second, third, and fourth sets of knotted loops.

**Discussion.** With appropriate training, novices can construct knots as strong and secure as experienced surgeons. A large percentage of knotted loops are insecure under testing conditions; extra throws may be required when using large diameter monofilament nylon. Novices may require extra training in maintenance of loop security when constructing square knots. Tying more than five or 10 knotted loops within a single training session does not provide added benefits for novices.

## INTRODUCTION

Knot tying is a key surgical skill in operative procedures (*Maturello et al., 2014*; *Muffly et al., 2011*; *Ching et al., 2013*). A secure knot is one that does not untie or slip to an extent that a vessel leaks or a wound opens (*Maturello et al., 2014*; *Thacker et al., 1975*). Knot security is affected by a variety of factors, such as suture size, material, number of throws, types of knots, and skill of the operator (*Muffly et al., 2011*; *Tera & Aberg, 1977*; *Tera & Aberg,*

Corresponding author
Karen Tobias, ktobias@utk.edu

*1976*; *Riboh et al., 2012*; *Avoine et al., 2016*; *Gillen et al., 2016*; *Ian et al., 2004*; *Trimbos & Klopper, 1985*; *Muffly et al., 2012*).

Because knot tying is a fundamental surgical skill, the construction of square knots using instrument and hand ties is taught to students and other trainees in basic surgical courses (*Thomas, Hayes & Demetriou, 2015*). Repetition is expected to improve knot tying skills and quality of knots produced by novice trainees, but fatigue during the learning process may negatively affect performance (*Thomas, Hayes & Demetriou, 2015*; *Compton et al., 2019*; *Asadayoobi, Jaber & Taghipour, 2021*). Instructors do not know how much practice is needed to solidify knot tying skills so that strong, secure knots are tied consistently.

The purposes of this study was to determine whether quality of hand-tied, square knots produced by novice trainees improved with repetition during a single knot tying session and to compare knot quality of trainees and experienced surgeons.

## MATERIAL AND METHODS

### Participants

This prospective study was approved by the University of Tennessee Institutional Review Board (UTK IRB-20-05723-XP). First year veterinary students enrolled in an elective, pass-fail surgery course were recruited as novice trainees for the study. Two surgeons certified by the American College of Veterinary Surgeons also participated as expert control subjects; both had more than 15 years of surgical experience.

### Suture materials

A 500-yard (457.2 meter) length of 30-pound (13.6 kg) test, 0.022 inch (0.55 mm) diameter nylon monofilament fishing line (Suffix Superior, Rapala VMC Corp., 17200 Vaaksy, Finland) was cut into fifty 30-foot-long pieces. Each piece was cut into 20 sutures 45.7 cm (18 inches) in length. One random suture from each piece was collected and placed into a bag to produce 50 bags that each contained 20 sutures representative of the entire 500 yards of monofilament line.

### Tying apparatus

Tying stands were fashioned from 1.5 cm (5/8 inch) diameter, polyethylene pipe (SharkBite PEX pipe, Cullman, Alabama, USA) and four cm (1 5/8 inch) diameter pipe insulation (Everbilt pipe insulation, Armacell, Chapel Hill, NC, USA). The pipe was cut into 30.5 cm (12 inch) long pieces, each of which was labelled with a number. The pipe insulation was cut into three cm (1.2 inch) long segments, and a segment was positioned near each end of the pipe to elevate the pipe off of the work surface. Each pipe was positioned so that the numbered end was toward the participant's left. Pipes were secured to work surfaces with a rope that ran through the pipe and was tied under the surface. Additionally, the ends of the pipes were taped to the work surface to prevent rolling or shifting.

### Trainee instruction

Each trainee was seated at a work station, and the trainee's name and pipe number were recorded. Each trainee was given a bag of 20 sutures, a pair of suture scissors, and a strand

of string as a practice suture. Before the experiment began, a board-certified surgeon (KMT) instructed the trainees on hand ties as part of the elective course. All trainees were taught to hold the excess suture in their left hand and use their right hand to pass and release the free end of the suture. Verbal instructions were illustrated with a PowerPoint presentation that included an overview of hand ties and knots; a video of a two-handed, hand-tied loop starting with a surgeon's throw and followed by three square throws (a 2=1=1=1 configuration (*Muffly et al., 2010*)); and then step-by-step instruction with word descriptions, photos, and videos of each step. During the step-by-step instruction, trainees performed each specific step as the video segment played, then paused until the next step was introduced. At the end of the presentation, the full video was played as a repeated loop, and the trainees practiced hand ties for 10 min using the provided string. The instructor was available for input during this time, but proficiency was not judged at this point. The trainees then cut and removed the string knots from their pipes.

## Knot tying

Trainees used the prepared 18-inch-long sutures to hand tie twenty 2=1=1=1 knotted loops (one knotted loop per suture), starting on the left side of the pipe and proceeding sequentially to the right. Trainees were asked to cut the suture ends, leaving them at least 5 mm in length. Trainees were given up to 50 min to complete the task. Once the trainee had completed the knotted loops, the rope ends were untied from the work surface and tied together over the top of the pipe and pipe insulation to prevent loss of any knotted suture loops. Two surgeons (KMT and PYM) also hand-tied twenty 2=1=1=1 knotted loops using the same materials as the trainees, except that the pipes were secured to a work bench that was at standing height, since both surgeons routinely performed surgery while standing.

## Informed consent

Once tying was complete, trainees were asked if they would be willing to submit their materials for a study testing the strength and security of hand-tied, knotted loops. Trainees were informed participation was voluntary and would have no effect on grade and that trainee names would not be associated with pipe numbers during testing. Trainees who were interested in participating placed their secured pipes in a plastic container that was sealed and stored in a temperature-controlled room until testing. Subsequent to course grade submission, an e-mail survey detailing the project and requesting electronic consent was sent to each trainee who had submitted materials for testing so that trainees could privately confirm or deny participation in the study.

## Testing

Testing was performed with a mechanical testing machine (Instron 5965 dual column table top AVE Extensometer model 2663-901, Instron, Norwood, Massachusetts) consistent with ASTM standard D2256 (*Yalcin, 2021*). To limit environmental and operator effects, all mechanical testing was performed on the same day by the same investigator (AGD). Before it was tested, each knotted loop was carefully slid off the pipe without handling the knots or pulling on suture ends. For each specimen, the knotted loop was positioned around two mechanical arms with the knot situated to the right, halfway between the two

arms. The loop was gently stretched at five mm/ minute to a preload of 5 N of tension to prevent it from slipping or shifting. The change in loop length during knot consolidation, known as extension under preload, was recorded. The machine was zeroed at the new length and tension, and a single test to failure was then performed: the loop was stretched at a rate of 100 mm/minute until it opened. The maximum load and the maximum change in length (extension) before loop failure were recorded.

For statistical purposes, each loop was categorized as one of four types based on its mode of failure: 1. Broke at knot—the suture broke at or near the knot without any slippage of the knot or loosening of the loop; 2. Slipped then broke—the knot slipped during testing as the throws tightened down to each other, but tension/load remained at $\geq 60\%$ of maximum and the loop broke under tension; 3. Slipped—a portion of the knot remained intact, but the loop loosened enough to produce a decrease in tension/load of $\geq 40\%$, ending the test; and 4. Untied—the knot completely untied and the loop opened, ending the test (*Maturello et al., 2014*; *Avoine et al., 2016*; *Drabble et al., 2021*). Mode of failure was based on the shape of the load–displacement curves (Figs. 1, 2, 3 and 4) and visual inspection of knotted loops (*Avoine et al., 2016*). Data from any sutures that failed during stretching under preload would be excluded from initial analysis. Any preloaded, knotted loops that failed when $\leq$5N of load was applied during single test to failure were categorized as "dangerous" (at risk for failure in a clinical situation) based on previous studies (*Ching et al., 2013*; *Drabble et al., 2021*; *Ind, Shelton & Shepherd, 2001*). For statistical purposes, knots also were categorized as secure (broke at knot or slipped then broke) or insecure (slipped or untied) based on the type of failure (*Babetty, Sumer & Altintas, 1998*; *Wasik, Cross & Voss, 2013*). A previously published power analysis indicated 10 knotted loops were required in each arm to detect a 5 N difference and a minimum of 17 knotted loops were needed to detect a temporal difference while maintaining a power of 80% with a type I error rate of 5% (*Muffly et al., 2012*; *Ind, Shelton & Shepherd, 2001*).

The difference between surgeons and trainees as well as the differences among modes of failure on extension at preload, maximum extension at failure, and maximum load to failure of knotted loops were evaluated using mixed model analysis for randomized block design. Diagnostic analysis on residuals was conducted to verify the normality and equal variance assumptions using Shapiro–Wilk test and Levene's test. The difference between surgeons and trainees in the number of each type of knot tied, as well as the differences among mode of failure depending on the temporal group (*e.g.*, first and last ten loops; or first, second, third, and fourth set of five loops), were analyzed using Poisson regression. Post hoc multiple comparisons were performed with Tukey's adjustment. Statistical significance was identified at the level of 0.05. All analyses were conducted in SAS 9.4 TS1M7 (SAS institute Inc., Cary, NC).

## RESULTS

Of the 39 novice trainees in the course, 30 completed 20 hand-tied, knotted loops (600 knotted loops in total) and consented to participate in the study. Two surgeons each completed 20 hand-tied, knotted loops (40 knotted loops in total).

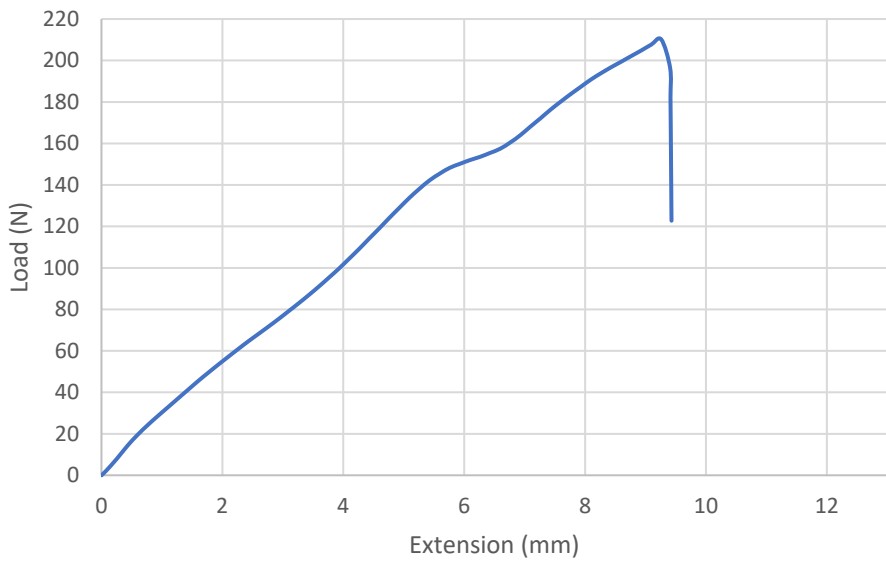

**Figure 1 Load-displacement curve for a loop that broke at the knot.** The blue line indicates the load displacement curve for a single knotted loop that broke at the knot. Note the gradual steady increase in load with extension until the sudden drop-off at the moment of breakage.

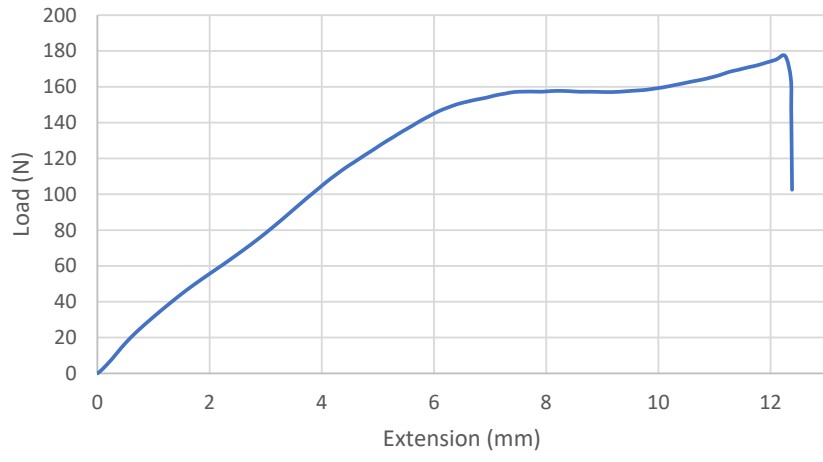

**Figure 2 Load-displacement curve for a loop that slipped and then broke at the knot.** The blue line indicates the load displacement curve for a single knotted loop that slipped before breaking at the knot. Note the gradual steady increase in load with extension, a plateauing of load with further extension, then a slight increase in load before the sudden drop-off at the moment of breakage.

## Modes of failure

All 640 knotted loops tolerated a preload of 5 N; thus, none were excluded from data analysis. Based on the modes of failure (Tables 1 and 2), trainees tied 140 secure knots (23.3%) and 460 insecure knots (76.6%), and surgeons tied 11 secure knots (27.5%) and 29 insecure knots (72.5%).

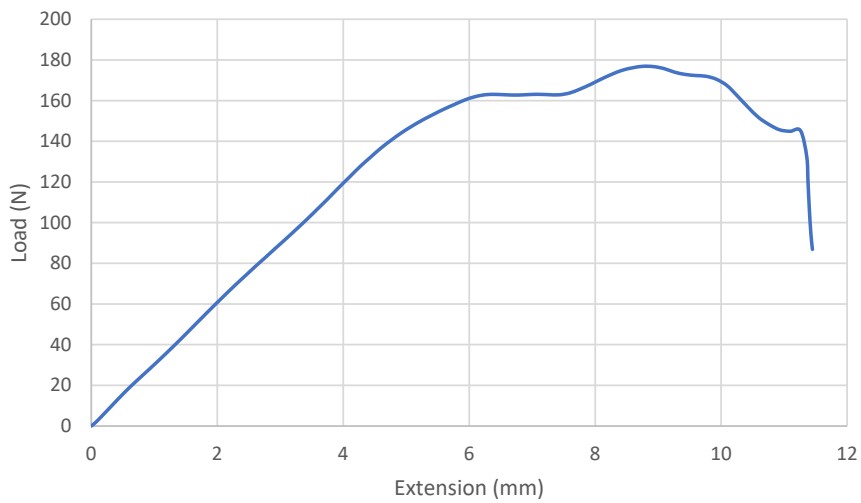

**Figure 3  Load-displacement curve for a loop that slipped.** The blue line indicates the load displacement curve for a single knotted loop that slipped without breaking. Note the gradual steady increase in load with extension, a plateauing of load with further extension, then a slight increase and then decrease as further extension is applied. The knots in this loop eventually came apart as one suture end slipped over the other, resulting in the sudden, rapid decrease in load.

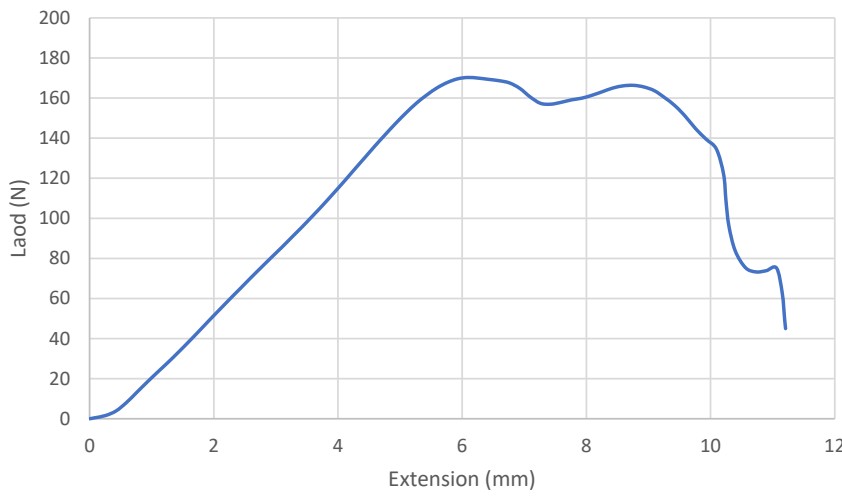

**Figure 4  Load-displacement curve for a loop that untied.** The blue line indicates the load displacement curve for a single knotted loop that untied without breaking. Note the gradual steady increase in load with extension, a plateauing of load with further extension, then a gradual decrease in load with further extension until the knots all untie.

When all knotted loops were compared as two temporal groups (first and last ten knotted loops) or four temporal groups (first, second, third, and fourth set of five knotted loops), there were no significant differences in the numbers of secure and insecure knots or in the modes of failure of temporal groups. Within temporal groups there were no significant

**Table 1** **Mechanical testing results for novice trainees' knotted loops.** Data is from mechanical testing of 600 novices' knotted loops that were placed under 5 newtons (N) preload and then tested to failure using an extension rate of 100 mm/minute. Results of extension at 5 N preload and maximum extension and load at failure are listed as means and standard deviations.

| Mode of failure | Extension at preload (SD), mm | Maximum extension at failure (SD), mm | Maximum Load to failure (SD), N |
|---|---|---|---|
| Broke at knot (51) | 1.1794 (0.3512)[a] | 17.5648 (3.8256)[a] | 136.7446 (39.7353)[a] |
| Slipped then broke (89) | 1.1224 (0.2501)[a] | 20.4431 (3.4073)[bc] | 90.2069 (24.0004)[b] |
| Slipped (268) | 1.2259 (0.3668)[a] | 20.2738 (5.1609)[bcd] | 65.4174 (12.5327)[c] |
| Untied (192) | 1.2984 (0.3806)[a] | 22.8546 (6.1088)[bd] | 74.6974 (18.8476)[d] |

**Notes.**
Within columns, means with the same letters are not significantly different.

**Table 2** **Mechanical testing results for surgeons' knotted loops.** Data is from mechanical testing of 40 surgeons' knotted loops that were placed under 5 newtons (N) preload and then tested to failure using an extension rate of 100 mm/minute. Results of extension at 5 N preload and maximum extension and load at failure are listed as means and standard deviations.

| Mode of failure (number) | Extension at preload (SD), mm | Maximum extension at failure (SD), mm | Maximum load to failure (SD), N |
|---|---|---|---|
| Broke at Knot (3) | 0.7876 (0.0519)[a] | 15.0000 (1.9827)[a] | 141.2040 (40.4764)[a] |
| Slipped then Broke (8) | 0.8412 (0.06624)[a] | 16.8501 (2.1183)[a] | 96.0430 (24.5073)[bc] |
| Slipped (22) | 0.5707 (0.1595)[a] | 16.9477 (2.7313)[a] | 60.6502 (9.3129)[bd] |
| Untied (7) | 0.7337 (0.1326)[a] | 22.7319 (5.1099)[b] | 72.2551 (26.0908)[bcd] |

**Notes.**
Within columns, means with the same letters are not significantly different.

differences in modes of failure or numbers of secure and insecure knots for surgeons and trainees ($p > 0.44$ for all values).

## Results of mechanical testing

With single test to failure, knots of 54 loops were classified as dangerous, including 53/600 (8.8%) of trainee knotted loops and 1/40 (2.5%) surgeon knotted loops ($p = 0.16$). All dangerous knots failed by untying. For trainees, 29 dangerous knots occurred in the first ten knotted loops, and 24 occurred in the second ten ($p = 0.47$).

Overall, knotted loops tied by surgeons had less extension under preload than those tied by trainees ($p = 0.0269$). Among trainee and surgeon groups, the amount by which the loop extended under preload was not associated with the mode of failure of the loop (*e.g.*, breaking, slipping, or untying; $p > 0.89$), its knot category (secure, insecure, or dangerous; $p \geq 0.88$) or whether the specimen was one of the first ten or last ten loops tied ($p = 0.2738$).

Overall, there was no difference between surgeons and trainees in terms of the maximum amount that loops extended before failure ($p = 0.3379$) or the maximum load measured before the loops failed ($p = 0.8827$). Mode of failure did have an effect on maximum extension and maximum load, however (Tables 1 and 2). Among both groups of participants, loops tied with secure knots tolerated greater loads ($p < 0.0001$) and extended more ($p \leq 0.0331$) before failure than those tied with insecure knots. For both groups, loops that broke at the knot tolerated the greatest load before failing ($p \leq 0.0104$), and those that untied had the greatest amount of extension before failing ($p \leq 0.0264$).

## DISCUSSION

Based on the number of insecure or dangerous knots and modes of failure, trainees did not show improvement in knot quality from their first to their last five or ten knotted loops. There was also no difference in the amount that loops extended under preload when comparing trainees' first and last ten knotted loops, indicating that the tightness or quality of their knots and loop security did not change with repetition. Results of trainees and surgeons were similar with regard to percentages of dangerous knots, modes of failure, mean extension at failure, and maximum load to failure. Surgeons' knotted loops had lower mean extension with preload than trainees, however, indicating greater loop security (*Luenam, Koonalinthip & Kosiyatrakul, 2019*). With appropriate training, novices can construct knots as strong and secure as experienced surgeons, although a large percentage of knots for both groups are insecure under testing conditions. Novices may require extra training in maintenance of loop security when constructing square knots.

This model used commercial monofilament nylon fishing line, which has a diameter similar to USP 2 nylon and has been used as suture for stabilization of canine stifle joints (*Huber, Egger & James, 1999*; *Sicard, Hayashi & Manley, 2002*; *Tinga et al., 2021*). The material was chosen because it is inexpensive, strong, and of large enough diameter that trainees and researchers could easily examine knots for flaws. Large diameter monofilament can have more stiffness and memory and a lower coefficient of friction than smaller or multifilament material and thus may be more difficult to handle or to tie into secure knots (*Huber, Egger & James, 1999*).

For this study, knotted loops were constructed with an initial surgeon's throw to help maintain a small loop that was positioned against the cylinder (*Muffly et al., 2010*). Knotted loops made with a surgeon's throw and three simple, square throws (2=1=1=1) are stronger than those consisting of four simple square throws (1=1=1=1) (*Zhou et al., 2013*; *Silver et al., 2016*). Based on research by *Muffly et al. (2011)*, use of five throws in a 2=1=1=1=1 construct may have provided better knot security.

Knot ends were cut to five mm of length or more to reduce the risk of knot failure, which is more likely to occur when knot ends are cut to less than 3 mm length (*Muffly et al., 2009*). There is no significant difference in strength when knot ends of 2-0 and 0 synthetic suture are cut to 3 mm or 10 mm (*Muffly et al., 2009*; *Schaaf, Glyde & Day, 2009*). Effects of knot end length on fishing line is unknown, however.

Before testing to failure, a preload was placed on each loop to prevent suture relaxation and to determine if the loop circumference extended with knot consolidation (*Luenam, Koonalinthip & Kosiyatrakul, 2019*; *Huber, Egger & James, 1999*). The amount of extension under preload provides a measure of loop security (*Luenam, Koonalinthip & Kosiyatrakul, 2019*). Surgeon and trainee loops extended by a maximum of 0.79 mm and 1.30 mm with preloading, respectively, indicating an initial tightening of throws within an acceptable margin of slippage (<3 mm elongation) for both groups of participants and a greater degree of loop security for surgeons (*Wasik, Cross & Voss, 2013*; *Luenam, Koonalinthip & Kosiyatrakul, 2019*; *Wu et al., 2018*). Some extension of the loop circumference is expected with tightening when a surgeon's throw is used (*Zimmer et al., 1991*). If suture tension is
applied unevenly or is not maintained between throws, the resulting construction may include half hitches, which can form a slip knot, or gaps between the throws, which can produce an "air" knot (*Burkhart et al., 1998*). Throws of slip and air knots can shift when placed under tension. Novice trainees have more difficulty maintaining secure loops of small circumference when tying square knots than when placing half hitches (*Wu et al., 2018*).

In this study, 8.4% of knots were considered dangerous. Dangerous knots are those that fail to resist 5 N of tensile force during single test to failure, since that is the minimum effective strength for knots used to secure pressurized vessels (*Ind, Shelton & Shepherd, 2001*; *Nathanson, Natahnson & Cuschieri, 1991*; *Ching et al., 2013*). This is independent of any pretensioning. All of the loops in our study tolerated a slow application of 5 N of force during pretensioning, possibly because gradual application of tension on knot ends can reorganize throws into a more secure knot (*Romeo et al., 2018*). Published rates of dangerous knots tied by trainees range from 3.4% to 13.5% and vary with suture material and size and amount and quality of tuition (*Ind, Shelton & Shepherd, 2001*; *Ching et al., 2013*). Intensive, supervised training on knot tying can immediately reduce the rate of dangerous knot formation (*Ind, Shelton & Shepherd, 2001*). For experienced surgeons, 0%–2.5% of knots are reported to be dangerous, depending on the type of knot and suture size (*Muffly et al., 2012*; *Romeo et al., 2018*).

Quality of knots based on modes of failure and knot security were similar for surgeons and trainees. Previous studies of have reported that 87% of trainee knots were slip knots and that only 15% to 20% of surgeons used correct knot construction to form square knots (*Ching et al., 2013*; *Trimbos, 1984*; *Thacker et al., 1977*). Use of monofilament nylon commonly results in formation of slip knots or to some degree of square knot slippage (*Drabble et al., 2021*). Clinical relevance of knot classification as secure or insecure is unknown; in this study, mean maximum load to failure for surgeon and trainee knotted loops exceeded 60N. Some authors consider knots that tolerate >30N of force before opening or slipping to be clinically safe (*Romeo et al., 2018*). Based on biomechanical studies of human abdominal walls, materials that retain suture strength and resist tearing at values greater than 20N are recommended for hernia repairs (*Deekin & Lake, 2017*). These studies do not take into consideration *in vivo* environments, repetitive cycling, and effect of activities such a coughing, straining, or jumping (*Muffly et al., 2010*; *Romeo et al., 2018*). In this study, loops that broke at the knot tolerated the highest maximum load to failure and therefore would be preferable in clinicals situations; however, knots that untie or slip can still maintain knot holding strength. Addition of more throws may prevent untying, although at the risk of increasing knot bulk (*Zhou et al., 2013*; *Ching et al., 2013*).

Based on maximum load to failure, quality of knots were similar for surgeons and trainees. Previous studies found that mean tensile strength of knots was actually higher and knot-tying technique better for trainees than instructors (*Harato et al., 2021*; *Batra et al., 1993*).

As in previous studies, the strongest knots were those that broke at the knot (*Avoine et al., 2016*; *Wu et al., 2017*). Knotted loops that broke at the knot also had the least extension before breakage. It is likely that knots that slipped, untied, or slipped then broke had less

friction and more knot consolidation (tightening and reconfiguring of the knot itself) during testing than those that broke at the knot (*Drabble et al., 2021*; *Huber, Egger & James, 1999*). Adding a fifth throw would have improved the strength of knots that were hitched (*Wu et al., 2017*).

Other studies on surgical learning note that surgical skills improve with practice; however, that improvement is usually based on duration of each attempt rather than the mechanical quality of the product (*Compton et al., 2019*; *Asadayoobi, Jaber & Taghipour, 2021*; *Gilmer et al., 2015*). Improvement in hand tie scores was noticed over three practice periods when students practiced tying two-handed ties for 15 min intervals and then were filmed making three attempts at hand ties (*Thomas, Hayes & Demetriou, 2015*). Time to knot completion was not measured in this study because the instructors wanted trainees to focus on technique and knot appearance rather than speed.

One study limitation was the use of unsterilized, large diameter, commercial nylon instead of 2-0 or 3-0 nylon or absorbable monofilament, suture materials and sizes commonly used for securing catheters or hand ligating blood vessels in veterinary medical general practices. Trainees and surgeons may have found smaller diameter sutures easier to tie squarely and securely, and adding an additional throw could have made the knots more secure (*Muffly et al., 2011*). Additionally, knot end length was not standardized among participants, although minimum lengths were met with all loops. Individual trainee performance was also not assessed temporally; it is possible some students may have demonstrated either improvement or fatigue over time. Because participants were part of an elective surgical course, there may have been a selection bias toward novice trainees with a better hand skills or more interest in surgery; results may therefore not reflect the general population of novice students.

## CONCLUSION

With repetition, novice trainees showed no improvement in knot quality based on mode of failure, mean extension under preload or load to failure, maximum load to failure, or numbers of dangerous knots. Based on these factors, tying more than five or 10 knotted loops in one session does not provide added benefits for novices. Novices and surgeons often tie insecure knots when using large diameter, monofilament nylon, particularly when placing only four throws. Knot insecurity and mode of failure are demonstrable with mechanical testing; it may be beneficial to incorporate such testing into novice training sessions to provide trainees with objective data regarding their performance.

### Funding
The authors received no funding for this work.

### Competing Interests
The authors declare there are no competing interests.

## Author Contributions

- Karen Tobias conceived and designed the experiments, performed the experiments, analyzed the data, prepared figures and/or tables, authored or reviewed drafts of the article, and approved the final draft.
- Pierre-Yves Mulon conceived and designed the experiments, performed the experiments, prepared figures and/or tables, and approved the final draft.
- Alec Daniels conceived and designed the experiments, performed the experiments, prepared figures and/or tables, and approved the final draft.
- Xiaocun Sun conceived and designed the experiments, analyzed the data, authored or reviewed drafts of the article, and approved the final draft.

## Human Ethics

The following information was supplied relating to ethical approvals (*i.e.*, approving body and any reference numbers):

The University of Tennessee Knoxville Institutional Review granted approval of UTK IRB-20-05723-XP on 7/20/2020.

## Data Availability

The raw data is available in the Supplemental Files.

## Supplemental Information

Supplemental information for this article can be found online at http://dx.doi.org/10.7717/peerj.14106#supplemental-information.

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
