# Peer review of "Does quality of novice hand-tied square knots improve with repetition during a single training session?"

_PeerJ, doi:10.7717/peerj.14106_

## Round 0.1 · original submission · Minor Revisions

Thank you for an interesting article. The experimental design is sound but there is a need for fairly substantial re-writing of some areas to improve clarity and readability. However as the methodology is sound, I have suggested minor revisions rather than major. The suggested deadline is 21 days, but if you require longer please let us know.

·

Basic reporting

The authors have submitted a well written manuscript that passes nearly all of the areas evaluated for basic reporting. My only concern here is that a specific hypothesis was not stated in the introduction. If this is a requirement of the journal then it should be added and then statements added to the discussion as to whether or not their hypothesis was supported. Honestly, I would defer to the editor for this decision. I don't feel strongly that the article fails in this area just because a hypothesis is not formally stated.

Experimental design

No comment. I commend the authors on a well designed study in a subject area in need of more objective research.

Validity of the findings

Findings are well reported and conclusions are reasonable given the results. There are a number of areas which the authors could elaborate on or discuss further which I will mention in the next section of this review. Overall, I feel the manuscript passes this section.

Additional comments

This review is for the initial submission of a manuscript evaluating the effect of repetition on the quality of two-handed hand ties by novices in a single session.

This study asks an interesting question that has implications for how hand ties are taught to veterinary students. Overall, it is a well-designed study that appropriately evaluates the question being asked. However, there are a number of clarifications and corrections necessary before it is acceptable for publication.

This is the first review I have done for PeerJ and am not entirely with their format for the reviews so I have structure my comments below in a format similar to what I have done with other journals.

Abstract
Line 27- Should be stated that it was two-handed hand ties

Introduction
A concise, narrative is provided setting up the reasons for the study and provides a clear purpose of the study. No hypothesis is stated. Based on the reviewer guidelines the authors should consult with the editor and consider adding a hypothesis.

Materials and Methods
The study details are well described and could be easily replicated from the description. I do have several questions, for clarification.

Line 98- Did you confirm that the tie was being performed correctly by each student before having them move on to tying the 20 test knots? Ie. Did they have to demonstrate proficiency? Do you know that the students continued to tie correctly when performing the 20 knot assessment? Were the knots assessed for errors prior to testing?
Line 103- How long was the initial instruction/lecture? What was the total amount of time the students were engaged in the activity, including performing the 20 test loops?
Line 131- How did you remove the loops from the pipe on which they were tied? Could this affect the testing?
Line 136- The manuscript states preload was applied slowly. How slowly? Was this a feature of the machine or arbitrarily chosen to ease the loop into preload? Why was preload set at 5N?
Line 138- Why was the 100mm/minute extension rate chosen? I have seen rates ranging from 20mm/min to 1000mm/min for studies evaluating nylon fishing string. Could loading rate affect the load curves?
Line 151- This line mentions loops that failed at <5N. Was this the first 5N applied after preloading to the initial 5N load? So essentially the loop failed at <10N total load?
Line 175-178- How do you think the use of only 4 throws on this size of suture impacted mode of failure? Is there evidence to suggest the appropriate amount of throws for this size and type of material? Readers may be tempted to misinterpret this study as providing evidence that nylon suture results in an insecure knot.
Line 195-196- The lines mention no difference in trainee’s mean extension for their first 10 and last 10 loops. Surgeons are not specifically mentioned in this sentence, but they are mentioned in all previous sentences of this paragraph. Was the mean extension at preload evaluated for the surgeons?
Line 232- Reference 23 recommended against using a surgeon’s throw with nylon fishing line because the surgeons knot resulted in decreased loop stiffness. Could this have affected your results?
Line 242-243- While I agree with the authors’ assessment about knot end length needing to be greater than 3mm I feel it is important to be clear about the comparisons and assessments that are being made. References 28 and 29 were evaluating 0 & 3 polydioxanone, 0 polygalactin 910, and 0 silk, not “nearly #2” nylon fishing line. As the authors pointed out in the previous paragraph, fishing line is different than suture and could affect the knots. This difference needs to be clearly stated to avoid confusing readers, at least the fact that the referenced 2 studies were looking at suture material and not fishing line.
Lines 283-288- Was this the case in your study? While the facts written in this paragraph are interesting, it does not appear that attention to technique by the participants was assessed in this study and I am having a hard time seeing how these statements add anything to the manuscript.
Lines 291-299- This sequence of sentences is a bit confusing. I think the authors might be trying to explain the finding in line 292 but I am having trouble connecting the relevance of some statements, particularly later in the paragraph. I would suggest that they revise the paragraph to better convey their thoughts.
Line 311-312- Prior ability and selection bias of the students could also be a factor. This was an elective course, correct? So enrolled students would presumably be more interested in the content and potentially have had some prior experience to drive that interest which could affect their performance. While it would be more challenging to do, it would be interesting to repeat this study using an entire class size. It has been my observation (anecdotal only) that students enrolled in skills electives generally are more interested and pick up the skill faster than their cohorts who are required to learn a skill in a core class. I regularly observe students in core classes who struggle to successfully complete hand ties after a single session of instruction and practice.
Line 316-322- How might your findings apply to or affect the teaching of hand ties and assessment of student learning?
Figure 2. It appears that this figure is named incorrectly.

·

Basic reporting

Thank you for asking me to review this paper.

The article structure was sound, and there are sufficient literature references.

However, the article was difficult to follow. I had to read it 5 times before I found that the information, as presented, “flowed”. Much of the language and terminology felt unnecessarily technical. It would have helped to provide explanations of “preload” and “single test to failure” at the start of the article. The description used in line 244 would have helped a reader to understand the article if it had been used at the start of the article!

I would advise using a simpler explanation in lines 136-139, such as “each loop was gently stretched till it was subjected to a force of 5N. The length by which each loop was further stretched was measured. The loop was then stretched at a rate of 100mm/min till the knot failed. The maximum load applied to each loop, and maximum length of extension of the loop, at the time of knot failure were recorded”, followed by a simple description of your four types of failure.

I wa somewhat confused by your description of knot failure as a decrease in tension > 40%. Please could you clarify this so it is easier to understand. Similarly, in lines 193-195, I would prefer to see an explanation of “mode of failure had no effect on extension at 5N”. Similarly, the term “extension at failure “ might be better understood by simpler terms as “ the amount by which the loop extended when the knot failed”

Lines 198-205 felt like a battering ram of comparisons! I think a better explanation of each result would clarify results for a reader,

The title of figure 2 on page 22 is wrong - it should be for knots that slipped and broke.

Experimental design

The design, and explanation of how each knot was tied, materials used, and how trainees were recruited, was thorough and detailed. The description of how each loop was tested was detailed, but confusing, till I saw explanation of ore-tension on line 244!

Validity of the findings

Main findings were valid; there was no improvement in knot quality comparing knots tied at the start of the session, and those tied at the end of the session. However, it could be argued that the number of knots tied by each participant- 20 - would be too small a number to allow improvement by repetition.

It could also be argued that more senior surgeons should have tied knots; comparing results from only 2 senior surgeons with those from 30 trainees involves comparing one very small group with a larger group. However, I don’t feel that this would markedly affect the validity of observations gained

The authors were courageous, and honest, in reporting that such large proportions of the knots tied did fail, and this my be related to the size and type of material used in the study.

However, I would advise limiting any discussion to results on]brained during the course of the study; the suggestion in the initial discussion in line 40-41 that extra training could improve the quality of knot tying is not borne out by your own data!

I was confused by your comment that 8.8% of knots tied by trainees and 2.5% of those tied by senior surgeons(1) were deemed as dangerous. At the start of your article you stated that any knot that could not withstand 5N of force was discounted from analysis, and in lines 149-151, yet explanation in lines 187-189 and especially in line 256 would suggest these “dangerous” knots would not cope with 5N of tension. This is confusing and needs clarification. Did they slip at 5N or not? The curve in figure 4 would suggest they did not.

Additional comments

Thank you for the article, the detailed description of your methodology, and your honesty with the proportions of knots that slipped! The article is useful, but was somewhat difficult to follow. The language used to describe how each knot was tested, and description of results, felt technical and I felt the whole article did not “flow” when I first read it. I do apologise!

I have made some specific suggestions regarding the language used, and would suggest explaining some of your definitions at the beginning of the article, to help the reader follow your article more easily from the start. I hope you don’t mind my suggestions.

---

## Round 0.2 · accepted · Accept

Thank you for your revisions. Having assessed the reviewer comments and responses from the authors, I am happy that the concerns have been addressed.
The re-writing for clarity is very welcome; I anticipate that the audience for this article will be educators as well as surgeons, and some veterinary educators may not have such a technical surgical background, so this is helpful.